# Interleukin-18 and Gelsolin Are Associated with Acute Kidney Disease after Cardiac Catheterization

**DOI:** 10.3390/biom13030487

**Published:** 2023-03-06

**Authors:** Po-Yen Kuo, Kai-Fan Tsai, Po-Jung Wu, Pai-Chin Hsu, Chien-Hsing Wu, Wen-Chin Lee, Hsiu-Yu Fang, Chih-Yuan Fang, Sheng-Ying Chung, Yung-Lung Chen, Terry Ting-Yu Chiou

**Affiliations:** 1Division of Nephrology, Department of Internal Medicine, Kaohsiung Chang Gung Memorial Hospital and Chang Gung University College of Medicine, Kaohsiung 83301, Taiwan; 2Division of Cardiology, Department of Internal Medicine, Kaohsiung Chang Gung Memorial Hospital and Chang Gung University College of Medicine, Kaohsiung 83301, Taiwan; 3Chung Shan Medical University School of Medicine, Taichung 40201, Taiwan

**Keywords:** acute kidney disease, cardiac catheterization, gelsolin, IL-18, L-FABP, urinary renal biomarkers

## Abstract

Patients undergoing cardiac catheterization are at high risk of post-procedure acute kidney injury (AKI) and may experience persistent renal damage after an initial insult, a state known as acute kidney disease (AKD). However, the association between AKD and urinary renal biomarkers has not yet been evaluated in this population. We enrolled 94 patients who underwent elective cardiac catheterization to investigate patterns of urinary renal biomarkers and their associations with post-procedure AKD. Serial urinary renal biomarker levels were measured during pre-procedure, early post-procedure (12–24 h), and late post-procedure (7–10 days) periods. In our investigation, 42.55% of the enrolled patients developed AKD during the late post-procedure period. While the liver-type free-fatty-acid-binding protein level increased sharply during the early post-procedure period, it returned to baseline during the late post-procedure period. In contrast, interleukin-18 (IL-18) levels increased steadily during the post-procedure period. Early post-procedure ratios of IL-18 and gelsolin (GSN) were independently associated with subsequent AKD (odds ratio (95% confidence interval), 4.742 (1.523–14.759) for IL-18 ratio, *p* = 0.007; 1.812 (1.027–3.198) for GSN ratio, *p* = 0.040). In conclusion, post-procedure AKD is common and associated with early changes in urinary IL-18 and GSN in patients undergoing cardiac catheterization.

## 1. Introduction

Acute kidney injury (AKI) after exposure to an iodinated contrast medium is a common and problematic condition owing to the extensive utilization of contrast-requiring procedures such as computed tomography and angiography [1]. Contrast exposure is the third most common cause of AKI during hospitalization. In high-risk populations, the incidence of contrast-associated AKI is approximately 11–12% [2,3]. It is also associated with adverse clinical outcomes including chronic kidney disease (CKD), end-stage renal disease (ESRD), cardiovascular events, prolonged hospital stays, and mortality [4,5,6,7]. Among contrast-requiring procedures, cardiac catheterization is particularly associated with a high risk of post-procedure AKI, ranging between 1.9–49% in the literature [2,8]. Apart from contrast-related factors (such as direct agent toxicity or post-exposure renal vasoconstriction), procedure-specific (such as atheromatous embolism) and patient-specific (such as hypotension or pre-existing renal disease) factors also play important roles in renal damage after cardiac catheterization [9,10,11,12]. Thus, renal injury after cardiac catheterization warrants particular attention and subsequent clinical monitoring [7,13,14].

Recently, the concept of acute kidney disease (AKD) was established to define persistent kidney damage 7–90 days after an initial nephrotoxic insult [13,15,16]. As a critical phase between AKI and CKD, AKD may represent a window of opportunity for modifying the disease course and preventing further kidney injury [4,17,18]. Therefore, AKD evaluation and prediction are essential in post-AKI management. Biomarkers such as urinary albumin-creatinine ratio (UACr), neutrophil gelatinase-associated lipocalin (NGAL), kidney injury molecule-1 (KIM-1), insulin-like growth factor-binding protein-7 (IGFBP-7), gelsolin (GSN), liver-type free-fatty-acid-binding protein (L-FABP), and interleukin-18 (IL-18) have been regarded as early predictors of kidney injury [19,20,21,22]. With the availability of enzyme-linked immunosorbent assay (ELISA)-based methods for their detection in urine samples, these renal biomarkers can serve as feasible and noninvasive tools for serial assessment in the post-AKI period [23]. While most renal biomarker studies have focused on a single time point after nephrotoxic events, the longitudinal changes in urinary renal biomarkers after cardiac catheterization and their associations with AKD are yet to be elucidated.

In this prospective observational study involving patients undergoing cardiac catheterization, we evaluated the incidence of AKD, time-course patterns of urinary renal biomarkers, and associations between urinary renal biomarkers and the occurrence of AKD after the procedure.

## 2. Materials and Methods

### 2.1. Patient Enrollment

Patients undergoing cardiac catheterization were recruited from the cardiology ward of the Kaohsiung Chang Gung Memorial Hospital between August 2020 and May 2021. Inclusion criteria were as follows: (1) adult patients (≥20 years of age) undergoing elective cardiac catheterization; (2) patients who had at least one risk factor for renal injury after cardiac catheterization (i.e., ≥60 years of age, diabetes, pre-existing renal impairment, coronary artery disease, cerebrovascular disease, heart failure (HF), or malignancy); and (3) patients with non-dialysis-dependent status. Patients were excluded if they had an AKI event within one month before enrollment, advanced CKD (Kidney Disease Improving Global Outcomes (KDIGO) stage 4–5 or ESRD) [24], documented anaphylactic allergy to iodinated contrast medium, hemodynamic instability at enrollment, or previous intravenous contrast exposure within one month before enrollment. All the participants received an intravenous fluid infusion with physiological saline at 60 mL/h for 8 h before cardiac catheterization as per the hospital’s practice for AKI prevention. The contrast medium utilized in the procedure was Omnipaque^®^ 350 (GE HealthCare, Chicago, IL, USA), which is a non-ionic, low-osmolality contrast agent containing 755 mg of iohexol (equivalent to 350 mg of organic iodine) per mL of solution. The study protocol was approved by the Institutional Review Board and Ethics Committee of the Chang Gung Medical Foundation, Taipei, Taiwan (IRB Nos. 201802329B0 and 201902059B0). The study adhered to the principles of the Declaration of Helsinki. Written informed consent was obtained from all the participants.

### 2.2. Measurement of Urinary Biomarkers and Identification of Post-Procedure Renal Injury

To assess the patterns of urinary renal biomarkers and occurrence of renal injury after cardiac catheterization, pre-procedure (on the procedure day), early post-procedure (12–24 h after the procedure), and late post-procedure (7–10 days after the procedure) levels of urinary renal biomarkers and serum creatinine (Cr) were measured in the study population. Urinary renal biomarkers including UACr, urinary IL-18, urinary GSN, and urinary L-FABP were measured in first-void spot urine samples in the morning using ELISA-based methods. The measurement of UACr was performed in the certificated laboratory of the hospital, and the measurement of other urinary biomarkers was performed by an experienced technician using commercial ELISA kits (IL-18: RayBio^®^ Human IL-18 ELISA Kit (ELH-IL18), RayBiotech Life, Inc., Peachtree Corners, GA, USA; GSN: ELISA Kit for Gelsolin (SEA372Hu), Wuhan USCN Business Co., Ltd., Wuhan, Hubei, China; L-FABP: NORUDIA L-FABP Kit (no. 84051000), Sekisui Medical Co., Ltd., Tokyo, Japan). The ELISA-based renal biomarkers were measured in accordance with the standard protocols of the manufacturers, and each result was corrected using the urinary Cr level of the same urine sample. A post-procedure AKI event was defined as an increase in serum Cr ≥26.53 μmol/L or ≥50% of the pre-procedure level during the early post-procedure period, according to the 2012 KDIGO criteria [25]. The post-procedure AKD was defined as an increase in serum Cr ≥26.53 μmol/L or ≥50% of the pre-procedure level or an increase in UACr ≥30% of the pre-procedure level during the late post-procedure period, which was based on the consensus of the Acute Dialysis Quality Initiative 16 Workgroup and previous studies regarding albuminuria as a surrogate marker of renal damage [15,26,27].

### 2.3. Collection of Baseline Demographic and Clinical Characteristics

Baseline demographic profiles of the enrolled patients were collected via the electronic medical record system of the hospital. Before the cardiac catheterization, we obtained information including age, sex, body mass index (BMI), smoking habit, pre-procedure blood pressure measurement, and comorbidities such as hypertension, diabetes, pre-existing renal impairment, coronary artery disease, cerebrovascular disease, HF, and malignancy history. Data on pre-existing renal impairment, including estimated glomerular filtration rate (eGFR) <60 mL/min/1.73 m^2^ and microalbuminuria (UACr ≥30 mg/g Cr), were recorded according to the data within three months before the enrollment. The contrast volume used during the cardiac catheterization was also collected from the hospital procedure records. Additionally, baseline hematological and biochemical profiles of the study population, including hemoglobin, hematocrit, glycated hemoglobin (HbA1c), lipid profiles, and eGFR levels, were measured before the cardiac catheterization. The Modification of Diet in Renal Disease equation, which is eGFR (mL/min/1.73 m^2^) = 175 × serum Cr (µmol/L)^−1.154^ × 0.0113 × age (year)^−0.203^ × 0.742 (if female), was used to retrieve the eGFR [28].

### 2.4. Statistical Analysis

Categorical variables are presented as numbers (*n*) with percentages, and continuous variables are presented as medians with interquartile ranges (IQRs) owing to the non-normal distribution revealed by the Kolmogorov–Smirnov method. The levels of serial urinary renal biomarkers (pre-procedure, early post-procedure, and late post-procedure) in the entire cohort were compared using the Wilcoxon signed-rank test. To evaluate the associations between urinary biomarkers and the occurrence of post-procedure AKD, we stratified the study population into two groups according to AKD occurrence during the late post-procedure period (AKD and non-AKD groups) and compared patient characteristics and serial urinary biomarker levels between the groups. Categorical variables were analyzed using the chi-square test, and continuous variables were analyzed using the Mann–Whitney U test for univariate analysis. Furthermore, the post-procedure ratios of all urinary biomarkers (defined as the ratio of the post-procedure level to the corresponding pre-procedure level) were also calculated during the early and late post-procedure periods and compared between the groups using the Mann–Whitney U test. To identify the factors independently associated with the occurrence of post-procedure AKD, urinary biomarker levels during the pre-procedure and early post-procedure periods and their post-procedure ratios during the early post-procedure period were assessed using multivariate logistic regression analysis via the forward stepwise selection method. Multivariate analysis was also adjusted for age, sex, BMI, pre-existing renal impairment, hypertension, diabetes, and baseline covariates with a *p*-value of <0.1 in the univariate analyses using the enter method. Statistical significance was set at a *p*-value of <0.05. Statistical Product and Service Solutions software (version 22.0; IBM, Armonk, NY, USA) was used for all the analyses.

## 3. Results

### 3.1. Characteristics of the Study Population

During the study period, 94 patients undergoing elective cardiac catheterization were enrolled for the analysis. The characteristics of the enrolled patients are summarized in Table 1. The median age of the study population was 66 years (IQR, 60–73), and 77.66% of the enrolled patients were males. The median BMI of the cohort was 25.55 kg/m^2^ (IQR, 23.20–28.31), and 15 (15.96%) of the participants were smokers. The median baseline eGFR level of the cohort was 76.33 mL/min/1.73 m^2^ (IQR, 60.33–96.21), and 19.15% and 44.68% of the participants had a baseline eGFR level of <60 mL/min/1.73 m^2^ and microalbuminuria, respectively. The most common comorbidity was hypertension (70.21%), followed by diabetes (38.30%), HF (19.15%), coronary artery disease (13.83%), cerebrovascular disease (5.32%), and malignancy (5.32%). The median contrast volume used in the procedure was 150.00 mL (IQR, 80.00–170.00). During the post-procedure period, none of the participants had repeated contrast exposure, nephrotoxic agent use, or shock episodes. The median serum Cr levels were 87.54 µmol/L (IQR, 69.85–111.63), 86.65 µmol/L (IQR, 64.10–109.42), and 92.84 µmol/L (IQR, 76.93–122.02) during the pre-procedure, early post-procedure (12–24 h), and late post-procedure (7–10 days) periods, respectively. During the early post-procedure period, 1.06% of the enrolled patients experienced an AKI event, and 40 (42.55%) participants had AKD during the late post-procedure period. Among the AKD group, 11 participants had an increase in serum Cr ≥26.53 μmol/L or ≥50% of the baseline level, 28 had an elevation in UACr ≥30% of the baseline level, and 1 met both the serum Cr and UACr criteria for AKD diagnosis.

### 3.2. Patterns of Urinary Renal Biomarkers in the Total Cohort after Cardiac Catheterization

The serial levels of urinary renal biomarkers are shown in Figure 1. The urinary IL-18 levels increased markedly during the late post-procedure period compared with those during the pre-procedure (*p* = 0.001) and early post-procedure (*p* = 0.045) periods; however, the early post-procedure levels were similar with the baseline levels (*p* = 0.178) (median (IQR), pre-procedure vs. early post-procedure vs. late post-procedure, 26.10 (17.40–40.73) vs. 30.90 (18.25–49.10) vs. 32.65 (19.95–51.90) ng/g Cr). In 65.96% of patients, urinary IL-18 levels were elevated above the baseline levels during the late post-procedure period. Furthermore, compared to pre-procedure levels, urinary L-FABP levels increased remarkably (*p* < 0.001) during the early post-procedure period. However, the late post-procedure L-FABP levels declined significantly compared to those during the early post-procedure period (*p* < 0.001) and were similar with the baseline levels (*p* = 0.067) (median (IQR), pre-procedure vs. early post-procedure vs. late post-procedure, 3.19 (1.60–9.36) vs. 13.52 (7.04–30.89) vs. 2.81 (1.13–5.75) μg/g Cr). Most of the enrolled patients (90.43%) presented with a higher L-FABP level during the early post-procedure period than during the pre-procedure period, and the late post-procedure L-FABP levels were lower in 93.62% of the patients than in the early post-procedure period. In contrast, UACr and urinary GSN levels were similar between serial measurements in the total population.

### 3.3. Patient Characteristics and Urinary Biomarkers Stratified by the Occurrence of AKD

To assess the association between urinary renal biomarkers and the occurrence of post-procedure AKD, the study population was stratified into two groups: AKD (*n* = 40) and non-AKD (*n* = 54). Compared to the non-AKD group, the proportion of patients with HF was higher in the AKD group (AKD vs. non-AKD, 30.00% vs. 11.11%, *p* = 0.021), and patients in the AKD group received a lower contrast volume during the procedure (median (IQR), AKD vs. non-AKD, 100.00 (75.00–150.00) vs. 150.00 (100.00–200.00) mL, *p* = 0.008). Additionally, diabetes was slightly less common in the AKD group than in the non-AKD group (AKD vs. non-AKD, 27.50% vs. 46.30%, *p* = 0.064). Although the late post-procedure serum Cr level was higher in the AKD group (median (IQR), AKD vs. non-AKD, 111.41 (83.11–133.51) vs. 88.42 (73.17–108.31) µmol/L, *p* = 0.020), the pre-procedure and early post-procedure serum Cr levels were similar between the groups. Other baseline demographic and clinical profiles, including the proportions of pre-existing renal impairment, such as microalbuminuria and baseline eGFR < 60 mL/min/1.73 m^2^, were similar between the groups (Table 1).

Comparisons of serial urinary biomarkers between the groups are presented in Figure 2 and Figure 3. The AKD group had significantly higher urinary L-FABP levels during the early and late post-procedure periods (median (IQR), AKD vs. non-AKD, early post-procedure: 18.24 (8.61–37.79) vs. 10.83 (5.57–18.48) μg/g Cr, *p* = 0.017; late post-procedure: 3.94 (1.86–8.06) vs. 2.58 (0.82–4.63) μg/g Cr, *p* = 0.035). The post-procedure ratios of UACr, urinary IL-18, and urinary GSN were markedly higher in the AKD group during the early post-procedure period (median (IQR), AKD vs. non-AKD, UACr: 1.29 (1.00–1.77) vs. 0.87 (0.62–1.23), *p* = 0.001; IL-18:1.30 (0.89–1.79) vs. 0.91 (0.69–1.34), *p* = 0.013; GSN: 1.20 (0.76–1.92) vs. 0.71 (0.43–1.42), *p* = 0.008). During the late post-procedure period, the post-procedure ratios of UACr, urinary IL-18, and urinary GSN were persistently higher in the AKD group (median (IQR), AKD vs. non-AKD, UACr: 1.68 (1.10–2.62) vs. 0.74 (0.37–0.97), *p* < 0.001; IL-18:1.55 (1.07–2.29) vs. 1.09 (0.70–1.68), *p* = 0.018; GSN: 1.28 (0.69–2.41) vs. 0.70 (0.24–1.67), *p* = 0.007). In contrast, the post-procedure ratio of urinary L-FABP was only slightly higher in the AKD group during the late post-procedure period (median (IQR), AKD vs. non-AKD, 1.14 (0.48–1.61) vs. 0.71 (0.33–1.27), *p* = 0.092).

### 3.4. Urinary Biomarkers Independently Associated with the Occurrence of Post-Procedure AKD

All the urinary biomarker levels during the pre-procedure and early post-procedure periods and their ratios during the early post-procedure period were evaluated by multivariate logistic regression analysis using a forward stepwise selection method to identify factors independently associated with post-procedure AKD. With adjustment for age, sex, BMI, pre-existing renal impairment, hypertension, diabetes, and baseline covariates with a *p*-value of <0.1 in the univariate analyses via the enter method, our analysis demonstrated that the post-procedure ratios of urinary IL-18 and GSN during the early post-procedure period were independently associated with the occurrence of AKD after cardiac catheterization (odds ratio (OR) (95% confidence interval (CI)), 4.742 (1.523–14.759) for IL-18 ratio, *p* = 0.007; 1.812 (1.027–3.198) for GSN ratio, *p* = 0.040). Moreover, the BMI was inversely associated with post-procedure AKD (OR (95% CI), 0.837 (0.709–0.989) per kg/m^2^, *p* = 0.036), whereas HF was positively correlated with AKD after the cardiac catheterization (OR (95% CI), 6.521 (1.356–31.351), *p* = 0.019) (Table 2).

## 4. Discussion

In this study involving patients undergoing cardiac catheterization and at risk of post-procedure renal injury, we demonstrated that the incidence of AKD was 42.55% and that the early post-procedure ratios of urinary IL-18 and GSN were independently associated with subsequent AKD. Additionally, distinct time-course patterns of urinary IL-18 and L-FABP after cardiac catheterization were also revealed in our investigation. While L-FABP increased sharply during the early post-procedure period (12–24 h), its level dropped toward baseline during the late post-procedure period (7–10 days). In contrast, IL-18 levels increased steadily throughout the post-procedure periods. Compared with the low AKI incidence during the early post-procedure period, the AKD incidence was remarkably high during the late post-procedure period in our study. Of note, the high AKD incidence might be partly ascribed to the inclusion of both serum Cr and UACr criteria for AKD diagnosis in our study. Additionally, previous studies have revealed that the incidence of contrast-associated AKI might be relatively lower during the immediate post-exposure period (24–36 h), and contrast-associated renal damage could become more prominent 2–5 days after exposure [3,4,29]. Considering the surrogate role of albuminuria for kidney damage [30,31] and the time-course variations of renal injury after contrast exposure, our study emphasized the requirement for serial monitoring of serum Cr and albuminuria (or other reliable renal biomarkers) in patients undergoing cardiac catheterization, especially in the high-risk population.

In our analysis, early post-procedure ratios of urinary IL-18 and GSN were independently associated with AKD occurrence. As a proinflammatory cytokine expressed in renal tubular cells, IL-18 can activate macrophages during renal inflammation and may play a crucial role in the pathogenesis of CKD [32,33,34]. Furthermore, in a large cohort of patients undergoing coronary angiography, urinary IL-18 levels were prominently elevated in those with major kidney events, thereby supporting its utilization in detecting acute renal injury [35]. Similarly, the potential role of GSN as a novel renal biomarker has been proposed [36,37]. GSN is a multifunctional actin-regulatory protein that influences cellular motility [38,39]. Altered renal GSN expression may disrupt the actin cytoskeleton in podocytes and result in podocytopathy [40]. In addition, plasma GSN can modulate immune response during inflammatory processes and is associated with adverse outcomes in patients with CKD and various illnesses [22,38,41,42,43,44]. In an animal study of AKI, gentamicin-induced nephropathy was linked to elevated urinary GSN excretion [21]. Moreover, an association between plasma GSN and AKI occurrence has been demonstrated in patients with sepsis or those receiving cardiopulmonary bypass, indicating its role in facilitating AKI diagnosis and monitoring [36,37]. Despite current evidence suggesting the predictive roles of IL-18 and GSN in AKI occurrence, their utilization in the AKD period has not been investigated. The association between AKD and urinary levels of these renal biomarkers is also a topic of interest because urine measurements are more feasible for serial renal monitoring [23]. However, since the reference ranges for urinary IL-18 and GSN have not been established, their optimal applications in clinical and investigatory settings remain unknown. In our study, by calculating the ratios during the early post-procedure period, we found that early elevations in IL-18 and GSN after cardiac catheterization were independently associated with the occurrence of subsequent AKD. Additionally, the steady increase in urinary IL-18 levels during the post-procedure period might also indicate escalating renal inflammation after cardiac catheterization in our study population [32]. While there are several other renal biomarkers with the potential to predict acute renal damage in various situations, such as NGAL, KIM-1, and IGFBP-7, the clinical application of IL-18 and GSN is still under investigation [45]. Our analysis highlights the potential utility of urinary IL-18 and GSN in identifying persistent kidney injury among patients undergoing cardiac catheterization.

Although the levels and post-procedure ratios of urinary L-FABP were not independently associated with subsequent AKD, a distinct pattern of urinary L-FABP levels was observed after cardiac catheterization in our study population. As an endogenous antioxidant abundant in the proximal renal tubules, urinary L-FABP levels reflect tissue oxidative stress during renal ischemia [46,47]. Previous studies also revealed that increased urinary excretion of L-FABP correlated with renal deterioration in patients undergoing cardiac catheterization and those with CKD [48,49]. In our study, a sharp increase in urinary L-FABP levels during the early post-procedure period occurred in 90.43% of the enrolled patients, and these levels returned to baseline during the late post-procedure period. This phenomenon may implicate transient renal ischemia after cardiac catheterization, resulting in an almost universal elevation of intrarenal oxidative stress in the study population, even in those without subsequent AKD. Further investigations combining different renal biomarkers are warranted to clarify the clinical significance of changes in L-FABP patterns and elucidate the pathogenesis of AKD after cardiac catheterization.

Our analysis also revealed that HF was positively associated with post-procedure AKD, and baseline BMI was inversely correlated with AKD after cardiac catheterization. Pre-existing HF is a well-documented risk factor for contrast-associated AKI, and the severity of HF may also serve as a positive predictor of AKI after contrast exposure [12,50,51]. Although obesity is a risk factor for AKI in critically ill patients, previous studies addressing the relationship between BMI and AKI have revealed mixed results [52]. Additionally, higher BMI has been reported to be paradoxically associated with fewer adverse outcomes and mortality events in patients with AKI after cardiac catheterization [53]. These observations and our findings underscore the importance and complexity of patient-specific factors in AKD pathogenesis after cardiac catheterization. Notably, while the AKD group received a lower contrast volume and had slightly fewer patients with diabetes, contrast volume and diabetes were not independently associated with post-procedure AKD in our study. In an investigation by Nikolsky et al., each 100 mL increment in contrast volume resulted in a 30% increase in the AKI risk in patients receiving percutaneous coronary intervention [54]. However, the effect of contrast volume on the AKI risk after percutaneous coronary intervention appears to have been smaller than those of other factors in the Mehran risk score, such as HF and other comorbidities [12,50]. In our study, the difference in median contrast volume between the AKD and non-AKD groups was only 50 mL, which might attenuate its impact on the risk of post-procedure AKD. Furthermore, although diabetes has been considered a risk factor for AKI in several scenarios, its effect might be substantially altered by the status of glycemic control [55]. Evidence has indicated that an elevated HbA1c is associated with a higher risk of AKI after cardiac catheterization, and aggressive glycemic control could reduce the AKI risk in patients with CKD and diabetes [55,56,57]. In our study, both AKD and non-AKD groups demonstrated fair glycemic control (HbA1c, median (IQR), AKD vs. non-AKD, 6.05 (5.70–6.85) vs. 6.10 (5.60–6.70) %, *p* = 0.888), which might minimize the influence of diabetes in the AKD risk after cardiac catheterization. Our findings reflect the multifactorial nature of AKD after cardiac catheterization, which requires further comprehensive studies to clarify its underlying mechanisms and preventive strategies.

This study has certain limitations. Because the occurrence of AKI was defined according to serum Cr levels during the early post-procedure period (12–24 h), the incidence of post-procedure AKI might be underestimated in our investigation. Additionally, as we identified post-procedure AKD events based on late post-procedure levels (7–10 days) of serum Cr and UACr, the possible occurrence and progression of AKD between 11–90 days after cardiac catheterization were not assessed in this study. Although we recognized the association between urinary renal biomarkers and AKD after cardiac catheterization, the predictive potential of these biomarkers for AKD was not suitable to be evaluated in this study because of the sample size. Similarly, the roles of other patient-specific and procedure-specific factors (such as diabetes and contrast volume) on the risk of post-procedure AKD require additional research with a larger sample size. Finally, the correlations between post-procedure AKD and other promising novel renal biomarkers (such as KIM-1 and NGAL) in the enrolled patients were not evaluated due to the study design. Further comprehensive investigations are warranted to address these issues and to clarify the clinical roles of urinary IL-18 and GSN in predicting AKD after contrast exposure. Despite these limitations, our analysis highlights the high incidence of post-procedure AKD, distinct time-course patterns of urinary IL-18 and L-FABP, and associations between AKD occurrence and early post-procedure ratios of IL-18 and GSN in patients undergoing cardiac catheterization, which have not been investigated in the literature. Therefore, our study provides a foundation for future research to elucidate the utility of these renal biomarkers in patients undergoing cardiac catheterization, who have a particularly high risk of post-procedure renal damage.

## 5. Conclusions

In patients undergoing cardiac catheterization and at risk of post-procedure renal injury, post-procedure AKD is common and associated with early changes in urinary IL-18 and GSN. The steady elevation of urinary IL-18 and transient increase in urinary L-FABP levels during the post-procedure period might imply intrarenal inflammation and ischemia after cardiac catheterization. Recognizing the different patterns of urinary renal biomarkers in various kidney injury etiologies may offer new insights into their mechanisms and pathogenesis, leading to innovative diagnostic and therapeutic paradigms.

## Figures and Tables

**Figure 1 biomolecules-13-00487-f001:**
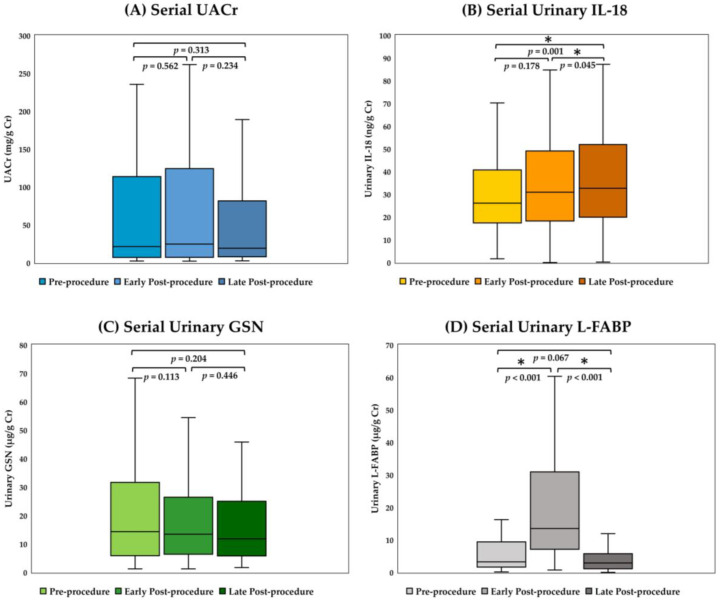
Patterns of Serial Urinary Biomarkers after Cardiac Catheterization. (**A**) Serial UACr; (**B**) serial urinary IL-18; (**C**) serial urinary GSN; (**D**) serial urinary L-FABP. The urinary biomarker levels during the pre-procedure, early post-procedure (12–24 h), and late post-procedure (7–10 days) periods were as follows (median (IQR)): UACr, 21.15 (6.93–113.68), 24.60 (7.10–124.30), and 19.10 (7.85–81.58) mg/g Cr; urinary IL-18, 26.10 (17.40–40.73), 30.90 (18.25–49.10), and 32.65 (19.95–51.90) ng/g Cr; urinary GSN, 14.30 (5.83–31.68), 13.40 (6.35–26.45), and 11.80 (5.75–25.00) μg/g Cr; urinary L-FABP, 3.19 (1.60–9.36), 13.52 (7.04–30.89), and 2.81 (1.13–5.75) μg/g Cr. Cr, creatinine; GSN, gelsolin; IL-18, interleukin-18; IQR, interquartile range; L-FABP, liver-type free-fatty-acid-binding protein; UACr, urinary albumin-creatinine ratio. * *p* < 0.05.

**Figure 2 biomolecules-13-00487-f002:**
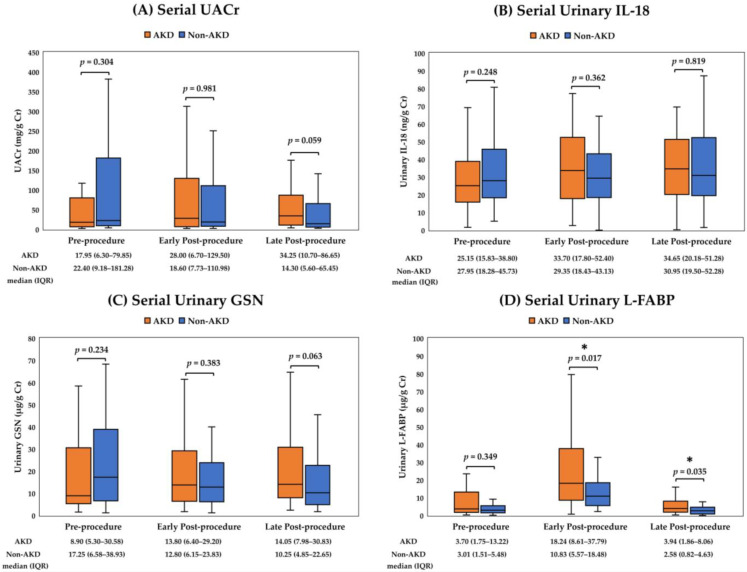
Comparisons of Serial Urinary Biomarkers between Groups. (**A**) Serial UACr; (**B**) serial urinary IL-18; (**C**) serial urinary GSN; (**D**) serial urinary L-FABP. AKD, acute kidney disease. * *p* < 0.05.

**Figure 3 biomolecules-13-00487-f003:**
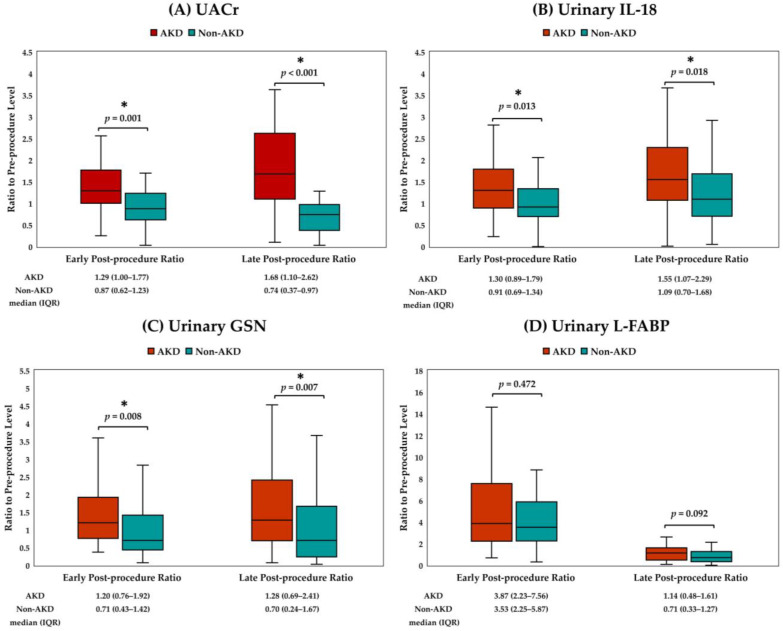
Comparisons of Post-procedure Ratios of Urinary Biomarkers between Groups. (**A**) UACr; (**B**) urinary IL-18; (**C**) urinary GSN; (**D**) urinary L-FABP. The post-procedure ratio of a urinary biomarker was defined as the ratio of the post-procedure level to the corresponding pre-procedure level. Post-procedure ratios of all the urinary biomarkers were calculated during the early (12–24 h) and late (7–10 days) post-procedure periods after the cardiac catheterization. * *p* < 0.05.

**Table 1 biomolecules-13-00487-t001:** Characteristics of the Enrolled Patients.

	Total (*n* = 94)	AKD (*n* = 40)	Non-AKD (*n* = 54)	*p*-Value
Age (year), median (IQR)	66 (60–73)	65 (58–77)	66 (61–72)	0.809
Male, *n* (%)	73 (77.66)	29 (72.50)	44 (81.48)	0.301
BMI (kg/m^2^), median (IQR)	25.55 (23.20–28.31)	25.69 (23.55–28.15)	25.47 (22.88–28.65)	0.979
Smoking, *n* (%)	15 (15.96)	8 (20.00)	7 (12.96)	0.357
Hypertension, *n* (%)	66 (70.21)	26 (65.00)	40 (74.07)	0.342
Diabetes, *n* (%)	36 (38.30)	11 (27.50)	25 (46.30)	0.064 ^#^
Coronary artery disease, *n* (%)	13 (13.83)	7 (17.50)	6 (11.11)	0.375
Cerebrovascular disease, *n* (%)	5 (5.32)	3 (7.50)	2 (3.70)	0.417
HF, *n* (%)	18 (19.15)	12 (30.00)	6 (11.11)	0.021 *
Malignancy, *n* (%)	5 (5.32)	3 (7.50)	2 (3.70)	0.417
eGFR < 60 mL/min/1.73 m^2^, *n* (%)	18 (19.15)	7 (17.50)	11 (20.37)	0.727
Microalbuminuria, *n* (%)	42 (44.68)	17 (42.50)	25 (46.30)	0.714
Systolic BP (mmHg), median (IQR)	131.50 (115.75–147.25)	126.00 (114.25–143.75)	134.00 (119.75–149.00)	0.146
Diastolic BP (mmHg), median (IQR)	72.00 (64.00–81.25)	71.00 (62.50–82.00)	72.00 (65.00–81.00)	0.671
Hemoglobin (g/L), median (IQR)	13.90 (12.30–14.73)	13.60 (12.30–14.78)	13.95 (12.45–14.75)	0.474
Hematocrit (%), median (IQR)	41.20 (36.95–43.88)	40.60 (36.73–43.55)	41.75 (37.30–44.28)	0.315
Triglyceride (mmol/L), median (IQR)	1.18 (0.91–1.74)	1.06 (0.89–1.82)	1.22 (0.94–1.74)	0.318
LDL (mmol/L), median (IQR)	2.31 (1.74–3.08)	2.36 (1.84–3.30)	2.25 (1.61–3.06)	0.502
HDL (mmol/L), median (IQR)	1.06 (0.93–1.27)	1.13 (0.96–1.31)	1.01 (0.87–1.25)	0.124
HbA1c (%), median (IQR)	6.10 (5.68–6.70)	6.05 (5.70–6.85)	6.10 (5.60–6.70)	0.888
Contrast volume (mL), median (IQR)	150.00 (80.00–170.00)	100.00 (75.00–150.00)	150.00 (100.00–200.00)	0.008 *
Serum Cr (µmol/L), median (IQR)				
Pre-procedure	87.54 (69.85–111.63)	95.05 (71.18–116.05)	85.77 (68.97–99.25)	0.149
Early post-procedure (12–24 h)	86.65 (64.10–109.42)	100.80 (63.44–119.81)	80.46 (64.10–99.25)	0.118
Late post-procedure (7–10 days)	92.84 (76.93–122.02)	111.41 (83.11–133.51)	88.42 (73.17–108.31)	0.020 *

AKD, acute kidney disease; BMI, body mass index; BP, blood pressure; Cr, creatinine; eGFR, estimated glomerular filtration rate; HbA1c, glycated hemoglobin; HDL, high-density lipoprotein cholesterol; HF, heart failure; IQR, interquartile range; LDL, low-density lipoprotein cholesterol; *n*, number. * *p* < 0.05; # *p* < 0.1.

**Table 2 biomolecules-13-00487-t002:** Factors Independently Associated with AKD after Cardiac Catheterization.

	OR	95% CI	*p*-Value
Age (year)	0.940	0.877–1.008	0.085
Male	0.402	0.086–1.877	0.247
BMI (kg/m^2^)	0.837	0.709–0.989	0.036 *
eGFR < 60 mL/min/1.73 m^2^	1.785	0.324–9.828	0.506
Microalbuminuria	0.589	0.168–2.067	0.409
Hypertension	1.212	0.316–4.650	0.780
Diabetes	0.818	0.230–2.918	0.757
HF	6.521	1.356–31.351	0.019 *
Contrast volume (mL)	0.993	0.984–1.003	0.200
Early post-procedure ratio of urinary IL-18	4.742	1.523–14.759	0.007 *
Early post-procedure ratio of urinary GSN	1.812	1.027–3.198	0.040 *

CI, confidence interval; GSN, gelsolin; IL-18, interleukin-18; OR, odds ratio. * *p* < 0.05.

## Data Availability

All data generated in this study are available from the corresponding author (tytc107@gmail.com) upon reasonable request owing to the hospital’s research regulations.

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
