# Peer review of "Interleukin-18 and Gelsolin Are Associated with Acute Kidney Disease after Cardiac Catheterization"

_biomolecules, 2023, doi:10.3390/biom13030487_

Round 1

Reviewer 1 Report

In this manuscript the authors report that increase of urinary interleukin-18 and gelsolin at 24 hours after contrast injection for cardiac angiography may predict acute kidney disease.

Abstract

Line 19 The statement “….urinary novel renal biomarkers were measured before, 24 h and 7 days after the procedure” and the description in the Methods section do not fully reflect what is stated in the Limitation paragraph of the Discussion section.

Methods:

The brand of radiocontrast used should be mentioned.

The authors cite the definitions of AKD as published the cited references, but give no description of how AKD was determined in their cases.

Statistical analysis: In instances such as in Table 2 and Figure 1, ANOVA (or its non-parametric equivalent) for repeated measures should be more appropriate than t-test.

Table 1:

What are those result numbers in parenthesis? Are they standard deviations? Are all the data normally distributed?

Figure 1, Tables 2 and Table 3

What is the difference between Table 2 and Figure 1? The same set of results should be only presented in one mode.

Table 3 is superfluous. It should be deleted.

Table 5

The selection of dependent and independent variables for the multivariate regression and the rationale for not using creatinine increment or ratio as the dependent variable should be explained.  Why are the pre-procedure and 24-hour Cr and urinary markers not included? Why does the regression only include the 24-hour to baseline ratios of albumin, IL-18 and GSN, but not L-FABP?

Figure 2

To justify the statements in the Conclusions section, further analysis for sensitivity, specificity and ROC performance is needed.

Author Response

Dear reviewer,

Thank you very much for your valuable comments and suggestions. They are extremely helpful in refining our manuscript. We have carefully addressed all the concerns and made corresponding revisions according to your advice. Please find our point-by-point responses to your comments and suggestions, which have been uploaded as a separated file (please see the attachment).

Reviewer 2 Report

In this manuscript, Dr. Kuo and colleagues measured the levels of several urinary factors within 24 hours and at 7-10 days after cardiac catheterization to find potential new urinary biomarkers for contrast-associated AKI or AKD. Their results suggested that the ratios of the 24-h post- procedure level to the basal level of IL-18 and gelsolin maybe independent factors associated with acute kidney disease. Considering that there have been fair numbers of clinical reports about the association of IL-18 level with contrast-associated AKI and AKD, and no significantly changed urine GSN level was detected at 24-h or 7 days post-procedure (Figure 1 and table 2), thus this study lacks the novelty and significance.

Other specific comments for the authors

1.       More background about gelsolin is needed. More explanations are needed to tell the readers why authors want to investigate the change of this protein in the contrast-associated AKI or AKD. A previous report found serum gelsolin level is associated with cardiac surgery induced AKI {PMID: 29427164}, why did the authors want to investigate gelsolin level in urine.

2.       Why the authors only measured urinary levels of IL-18 and gelsolin, how about their levels in serum?

3.       To better understand the urinary biomarkers for contrast-associated AKI or AKD, the authors should compare the specificity and sensitivity of these new biomarkers (IL-18 and gelsolin in this study) with some well-known biomarkers, such as KIM1 and NGAL.

Author Response

(The authors gave the same response as above.)

Round 2

Reviewer 1 Report

The manuscript is unnecessarily lengthy and frustratingly painful to read. The extensive revisions have made it worse. The texts are often redundant, repetitive, unfocused and even incoherent throughout the manuscript. It should be completely revamped.

As an example, the text between lines 39-45 is unnecessarily redundant and repetitive; it not only does not accurately represent the findings of the cited references, it even weakens the merit of the current study.

“Additionally, previous research has revealed that AKI after contrast exposure is associated with several adverse outcomes, including chronic kidney disease (CKD), end-stage renal disease (ESRD), cardiovascular events, and mortality [4,5]. Furthermore, regardless of whether clinically obvious AKI occurs in an immediate post-exposure phase, patients exposed to contrast medium may have increased risks of cardiovascular events, renal deterioration, and prolonged hospital stay [6,7].”

Another example is the term “changing ratio” throughout the text. A ratio is used to represent the relative magnitude of change. The word “changing” is redundant and confusing.

In Table 1, 27% of the AKD group and 46% of the non-AKD group had diabetes. This difference is not only paradoxical it is also not reflected in HbA1c data. This discrepancy should be explained.

The median contrast volume was 50% higher in the non-AKD group. This paradox suggests that the contrast volume administered had adjusted by the cardiologists for patient characteristics such as body weight and renal function. If true, the contrast volume should not be treated as an independent variable in the multivariate analysis. The issue of multicollinearity should be addressed.

The incidence of CA-AKD is much higher than the 11%-12% CA-AKI as cited in Introduction (line 38). The reasons of this discrepancy and its implication in the interpretation of the results should be addressed in the Discussion section.

The peculiar finding that diabetes and contrast volume are not independent risk factors of CA-AKD should be related to the findings in the literature.

Author Response

Dear reviewers,

Thank you very much for your valuable comments and suggestions. They are extremely helpful in refining our manuscript. We have carefully addressed all the concerns and made corresponding revisions according to your advice. Please find our point-by-point responses to your comments and suggestions, which have been uploaded as separated files. Additionally, one reviewer suggested extensive English editing, and the revised manuscript has been edited by the Editage to improve language accuracy, with the editing certificate uploaded as unpublished material.

We would like to thank you and reviewers for taking the time and effort necessary to review the manuscript, and we hope the revised manuscript is now in a form acceptable for publication. Thank you for your consideration and we look forward to your response.

Reviewer 2 Report

Authors have addressed my concerns.

Author Response

(The authors gave the same response as above.)
